# Novel artificial selection method improves function of simulated microbial communities

Björn Vessman[1☯], Pablo Guridi-Fernández[1,2☯], Flor Inés Arias-Sánchez[3], Sara Mitri[1,4]*

**1** Department of Fundamental Microbiology, University of Lausanne, Lausanne, Switzerland, **2** Instituto de Biomedicina y Biotecnología de Cantabria (University of Cantabria - CSIC), Santander, Spain, **3** BIH Center for Regenerative Therapies (BCRT), Charité - Universitätsmedizin Berlin, Berlin, Germany, **4** Swiss Institute of Bioinformatics, Lausanne, Switzerland

☯ These authors contributed equally to this work.
* sara.mitri@unil.ch

## Abstract

There is increasing interest in artificially selecting or breeding microbial communities, but experiments have reported modest success. Here, we develop computational models to simulate two previously known selection methods and compare them to a new "disassembly" method. We evaluate all three methods in their ability to find a community that could efficiently degrade toxins, whereby investment into degradation results in slower growth. Our disassembly method relies on repeatedly competing different communities of known species combinations against one another, while regularly shuffling around their species combinations. This approach allows many species combinations to be explored, thereby maintaining enough between-community diversity for selection to act on, and resulting in communities with high performance. Nevertheless, selection at the community level in our simulations did not counteract selection at the individual level, nor the communities' ecological dynamics. Species in our model evolved to invest less into community function and more into growth, but increased growth compensated for reduced investment, such that overall community performance was barely affected by within-species evolution. Within-community ecological dynamics were more of a challenge, as we could control them during the selection process, but community composition and function dropped in the longer term. Our work shows that the strength of disassembly lies mainly in its ability to explore different species combinations, and helps to propose alternative designs for community selection experiments.

## Author summary

Artificial selection has been extremely powerful in improving properties of complex biological or biochemical entities. The most familiar examples come from the breeding of animals and plants, but directed evolution has also been applied to

**Data availability statement:** https://github.com/Mitri-lab/artif_comm_select_model.

**Funding:** BV, FAS, PGF and SM were funded by European Research Council Starting Grant 715097, and SM by the NCCR Microbiomes (Schweizerischer Nationalfonds zur Förderung der Wissenschaftlichen Forschung), grant number 51NF40_180575. The funders had no role in study design, data collection and analysis, decision to publish, or preparation of the manuscript.

increasing the efficacy of enzymes. Microbial communities are the promising next frontier for artificial selection, as they have the potential to overcome many human problems, from efficiently degrading plastic to increasing agricultural yield. Yet, breeding stable, well-functioning communities is hard: published selection experiments from the last decade have shown only minor improvements over naturally assembled communities. Before conducting more experiments, models can help to better understand the problems with artificially selecting communities and how to improve current approaches. Here, we develop mathematical models that show one reason why previous selection methods have been unsuccessful, propose a new selection method that overcomes this problem and analyze its strengths, weaknesses and experimental feasibility. The essence of the new method is to explore many different species combinations in a way that gradually improves community performance to ultimately find the best species combinations. According to our models, our new method is expected to outperform previous methods significantly.

## Introduction

Humans have been breeding plants and animals for centuries by allowing individuals with the most desirable traits to selectively produce offspring. Also known as "artificial selection" or "directed evolution", breeding has altered traits such as the size of fruits or the enzymatic activity of proteins used in biotechnology [1]. More recently, we have started to appreciate that microbes — often multi-species communities of microbes — play an important role for health and the environment. One way to improve or optimize the functions and services that these microbes provide is to select for their traits in the same way as traditional breeding.

However, breeding microbial communities is less straightforward than individual organisms [2–6], mainly because the breeder selects whole groups of organisms rather than individual plants, animals or proteins. Heritability – "the regression slope of average offspring phenotype on average parental phenotype" [7] – is one of the requirements for evolution by natural selection, but groups of organisms naturally have lower heritability compared to canonical individuals [8]. The problem arises because a single community-level "generation", which we call a "round of selection" to avoid confusion, can comprise several generations of cells, each belonging to different genotypes (i.e. species or strains). Since the genotypes all reproduce at varying rates, their relative abundances can change during one round of community growth and over subsequent rounds. Because community function depends on the traits of all of its constituent genotypes and their proportions, an "offspring" community may not resemble its "parent" [8–10]. Another issue with community selection is that within- and between-species selection continue to operate within a round. If there are trade-offs between growth and contribution to the community trait, cheaters that contribute less can emerge and sweep to fixation [2,11]. A third challenge is to find a good constellation of different community members and their proportions that

can best achieve the desired function. Generating different constellations of member species at each round of selection is also important to have enough variability for selection to act on [8]. The major challenges for community-level selection then, are (i) ensuring that community functions are heritable, (ii) that within-community selection does not dominate over between-community selection, and (iii) ensuring variability, that communities differ in phenotype.

Since the earliest community breeding experiments by Swenson *et al.* where microbial communities were selected to increase plant yields and to control pH [12], many attempts have been made, aiming to optimize different microbial community traits [13,13–22]. Some of these studies have managed to significantly improve the *average* community function over several rounds of selection, but sometimes only as an effect of time without any significant differences between selection treatments [12,19,20]. Overall, community breeding experiments have shown mixed success [3,4,23], with computer simulations providing some clues on how to improve them [6,9,10,24–28].

Most previous experiments have followed one of two methods to propagate the communities with the highest scores to the next round: in the "propagule" selection method (PS), a fraction of the cells in the highest-scoring communities are selected and transferred by dilution, while in "migrant pool" selection (MS), all populations of the selected communities are mixed in a *pool* before they are diluted in equal proportions to the new tubes. These methods suffer from a rapid decrease in between-community variability [29], such that selection has little to act on. This suggests that we need novel selection methods that can better explore the search space of species combinations [26].

In this manuscript, we explore a selection method that we call "disassembly selection" (DS) that is designed to maintain between-community variability. After each round, we disassemble the selected communities by isolating the constituent species before recombining them into new communities for the next round of growth. Species are combined in equal proportions at every round to ensure that community traits are heritable. We use two computational models of evolving microbes in a well-mixed liquid culture to systematically compare our approach to the classical propagule selection (PS) and migrant pool selection (MS) methods. To inform the design of experiments to select for communities that degrade industrial pollutants [30,31], microbial communities in our simulations are selected for their ability to degrade toxins.

Our results confirm our intuition that propagule and migrant pool selection do not maintain enough variability to explore many different species combinations. In contrast, our disassembly approach maintains variability between communities, allowing it to find some of the best possible species combinations. Nevertheless, disassembly selection still suffers from an important problem: once our scaffold of restarting communities with equal abundances of each species is removed, communities lose their high function. Our work thereby shows that this new method can find species combinations whose community function is high, but leaves controlling the ecological dynamics of these successful communities as an open problem.

## Results

### Simulating community-level selection

To make sure that our results are not biased by the type of model we chose or its exact implementation, we developed an individual-based model (IBM) and a model based on ordinary differential equations (ODE). As both models showed similar results, we present the results of the IBM in the main text and leave the ODE model as a comparison in the supplement (Methods in S1 Note, statistics in S1–S3 Tables, parameter choices in S4 Table, results in S1, S3–S4, and S9 Figs). Each model has its strengths and weaknesses – the IBM being more flexible to implement experimental details and the ODE being simpler to reproduce and understand – but we chose to focus on one for conciseness and leave it to the reader to explore the other if preferred.

As these simulations were designed to inform a real artificial selection experiment, parameters were chosen to resemble the experimental setup [31]. Each simulation starts with 21 communities of 4 species each, chosen at random with replacement from a set of 15 initial species. We generated 5 such 15-species sets and repeated the simulations for each to make sure that our results are not specific to a given set of parameters. Each species is described by randomly drawn

model parameters: its growth and uptake rates for each of 4 available nutrients, its death and degradation rates for each of 10 toxic compounds, and a set of fractions $f$ representing its investment into the degradation of the 10 toxins. The remaining fraction $(1 - f \cdot$, shorthand for $1 - \sum f)$ of consumed nutrients can be invested into growth (see Methods). The microbes in our model thereby face a dilemma: whether to invest consumed nutrients into growth or into degradation of toxic compounds that would otherwise cause cell death.

In each of the 21 simulated "tubes", we place cells of different types, nutrients and toxic compounds (Fig 1A). At each time-step, with fixed probabilities (Table 1), a cell takes up nutrients that get invested into cell growth (activation and division) and mutation, or into toxin degradation. Mutations that occur only alter the investment fractions $f$, while all other species' parameters remain unchanged throughout the simulations. After 80 such time-steps, each of the 21 communities is scored based on degradation of the ten toxic compounds. The best 7 communities are then selected and propagated to the next round, depending on the selection method (Fig 1B).

## Propagation methods

In propagule selection (PS), the 7 communities with the highest score are diluted to make offspring communities, with the frequency of each offspring community corresponding to the score of their parent. In migrant pool selection (MS), the 7 best communities are combined and then all cells distributed amongst the 21 tubes of the next round (see Methods, Fig 1B).

In the disassembly method (DS), the 7 best parent communities are chosen based on their degradation score, but we also penalize communities if species extinctions occur. This is meant to favor more cooperative communities. To choose the species composition of each offspring, we replicate the best communities proportionally to their score, and randomly choose 5 offspring tubes to receive or lose a species, respectively. In some cases this can result in species exchange. When adding new species, we select them to ensure that each species is present in at least one of the 21 tubes at each round. For each of the chosen species, we place 10 cells (see Methods, Fig 1B, red box).

In addition to the selection treatments, we also implemented corresponding random controls (e.g. random propagule: PR) where 7 communities are chosen at random instead of according to their score, and a no-selection control (NS) where every community is diluted without selection (Fig 1B). This last control forms a baseline for how communities change due to species interactions [26,32], which can only occur through nutrient consumption and toxic compound degradation in our model. We ran 10 replicate simulations for each of the 5 species sets for each propagation method for 50 rounds of selection. The same initial conditions were used for the different selection methods to allow for a fair comparison.

## Disassembly selection finds communities whose degradation ranks in the top percentile of all possible communities

All simulated selection methods succeeded in improving the median degradation score across the 21 communities between round 0 and 50 (S2 Fig), which is consistent with previous work [3,26]. However, DS was the only propagation method to significantly and consistently improve the maximum degradation score, meaning that on average, the best community in round 50 degraded significantly better than the best community in round 0 (one-sided Wilcoxon signed rank-test $n = 50$, 10 repeated runs of 5 species sets, $p < 10^{-9}$, Fig 2A). The increase in maximum score in DS $(0.22 \pm 0.06)$, was also significantly different from the classical selection methods $(-0.03 \pm 0.06$ and $-0.12 \pm 0.09$ for PS and MS), from its own random control (DR), and from NS (all two-sided Wilcoxon tests of difference in max. degradation between DS and other methods, $n = 50$, $p < 10^{-9}$).

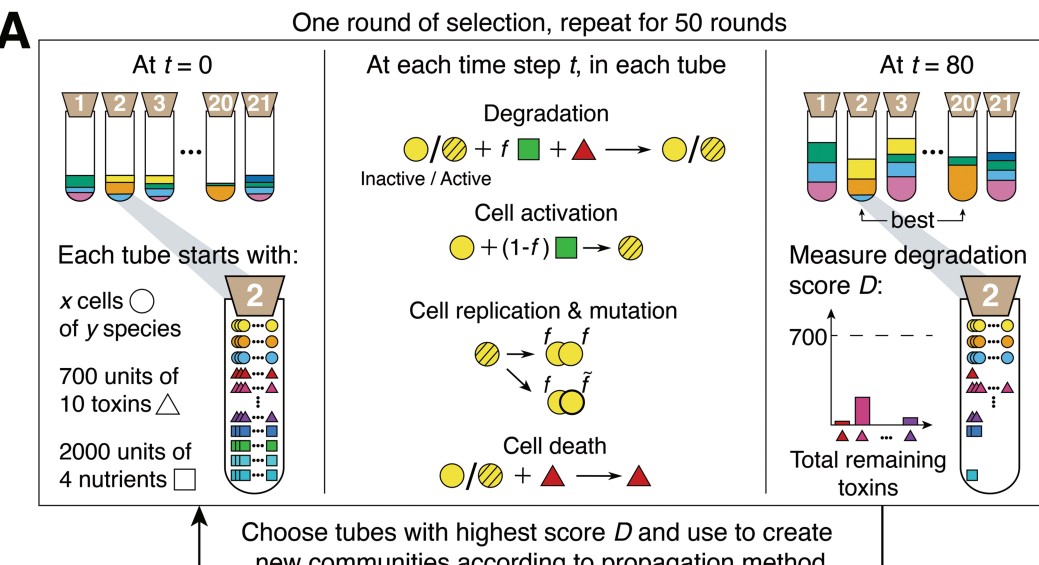

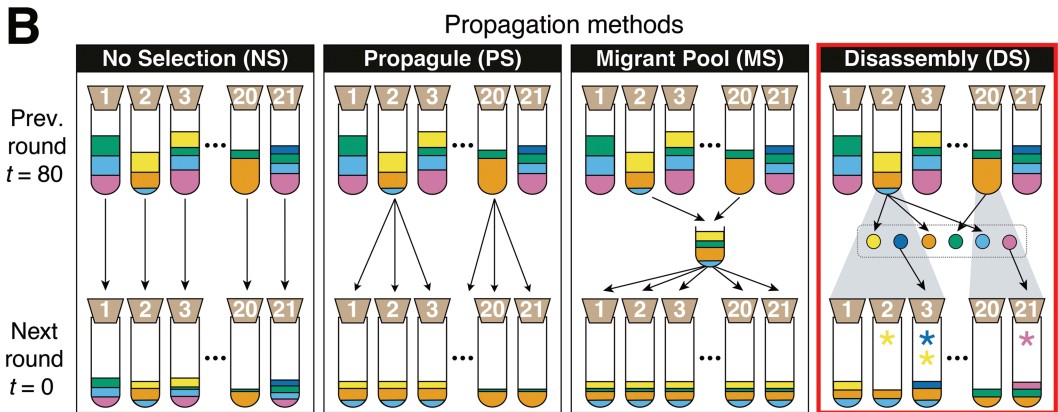

**Fig 1. (A) One round of artificial selection as implemented in the IBM (see S1 Fig for the ODE model).** Communities are illustrated as test tubes with bacterial "species" in different colors. At $t = 0$, each tube is inoculated with cells (circles), toxins (triangles) and nutrients (squares) as indicated. In disassembly selection, $x = 10$ cells of each species are used. We begin all communities with $y = 4$ species in round 0. At each time step $t$, each cell in each tube is subject to one of four processes (degradation, cell activation, cell division and mutation, and cell death) that occur with different probabilities (Table 1, see Methods for details). After 80 such time-steps, the degradation score $D$ is measured (see Equation 4), and the 7 communities with the highest score (illustrated with 2) are selected for propagation. In the random controls, 7 random communities are propagated. **(B) Propagation methods.** The 7 best communities (represented by tubes 2 and 20 here) are propagated to make the same number of communities for the next round. Propagule selection (PS): selected communities are diluted, contributing offspring communities in proportion to their degradation score. Migrant pool (MS): selected communities are merged and then diluted. Disassembly (DS), the method introduced in this paper, highlighted with a red border: Each species from the selected communities is isolated and saved in a repository (dotted rectangle). Each selected community contributes offspring communities in proportion to their degradation score (grey areas). A fraction of the new communities receive new species (red arrows) or lose members from the previous round (asterisk in color of removed or added species). No-selection (NS): As a control treatment, each community is diluted into a new tube. Propagule, migrant pool and disassembly have corresponding random treatments (PR, MR and DR), where community scores are ignored (see Methods).

As an absolute performance measure, we computed the degradation scores of all $2^{15} - 1 = 32767$ possible communities consisting of 1 up to 15 species for each species set and sorted them from best to worst. The communities found by DS ranked among the best few hundred, finding the very best community out of 32767 in 17 out of 50 runs (Fig 2C).

**Table 1**. **Parameters defining a microbial strain *i* in the IBM.** Growth rates, death rates and degradation investment vectors $r_{ij}$, $m_{ik}$ and $f_{ik}$ are made sparse by multiplying them by a vector drawn from Bernoulli(0.5). Each species can this way only take up a random fraction of nutrients, be affected by a random fraction of the toxic compounds and degrade another fraction of the toxic compounds. Despite changing by mutation, the total investment $f_{i.}$ is limited to the interval $[0,\ 1]$.

| Parameter | Description | Randomly sampled from |
|---|---|---|
| $l_i$ | ID of lineage $i$ | |
| $a_i$ | Activation probability | Beta(2,2) |
| $r_i$ | Replication probability | Beta(2,2) |
| $n_{ij}$ | Consumption rate of nutrient $j$ | Uni(0,1), Sparse, Rescaled so that $\sum_j n_{ij} = 1$ |
| $m_{ik}$ | Death rate of strain $i$ due to toxic compound $k$ | Uni(0.001,0.02), Sparse |
| $f_{ik}$ | Fraction of consumed nutrients invested into degradation of toxic compound $k$ | Uni(0,1), Sparse, Rescaled so that $\sum_k f_{ik} = Uni(0,1)$ |

## Communities selected by disassembly invest more into degradation and are composed of diverse species with complementary phenotypes

In our model, community performance depends on (a) the overall investment into degradation of toxic compounds relative to growth, and (b) how well community members complement each other. Community members will compete less if they take up different nutrients while the degradation score of a community can increase if its members specialize on degrading different toxic compounds (Equation 4).

To understand how these two properties changed over time, we first quantified the fraction $f_{.}$ of resources invested into degradation summed over all toxic compounds, averaged over the species in each community. Starting from an average investment of 0.5, DS finds communities that invest significantly more resources into degradation at round 50 than in the first round (one-sided Wilcoxon test, all $p < 10^{-9}$, $n = 50$, Fig 3A). This is not due to any single species with unusually high degradation capabilities, but rather because DS finds a combination of species with high investment. The average within-community species diversity increases over the 50 rounds (Fig 3C), which means that the communities consist of an increasing number of species and/or that the communities are increasingly even. Accordingly, in DS, the effective number of consumed nutrients and toxic compounds increases over the 50 rounds (Fig 3D and 3E). This increase in coverage and community diversity was not observed for the other selection methods (Figs 3C and S5).

Given the complementarity in nutrient uptake and toxic compound degradation, one might expect species to grow and degrade better together compared to when they are alone, as they may be facilitated by other species that remove toxic compounds that they themselves cannot. We use "synergy" to quantify whether a community property (e.g. degradation) is greater than that of its member species together (Fig 3F and 3G). Against a baseline of all possible species combinations for a given community size – richness in our models increases niche overlap and competition for resources, which decreases synergy – communities selected by DS have significantly higher synergy, for both degradation and cumulative biomass (Kruskal–Wallis *H* test, $p < 10^{-9}$, Fig 3F and 3G).

In sum, communities selected by DS invest more into degradation compared to communities from other methods. These communities are diverse in composition, consist of species with minimal niche overlap, and cover the toxic compounds evenly (Fig 3F).

## Disassembly can explore more species combinations by diversifying the selected communities

Seeing that communities selected by DS are diverse and efficient degraders, we now investigate how the method finds these communities. First, DS explores more species combinations than the other methods (Figs 4A and S7). The classical propagule method (PS) can only find sub-communities of the species combinations present in round 0. Similarly, while

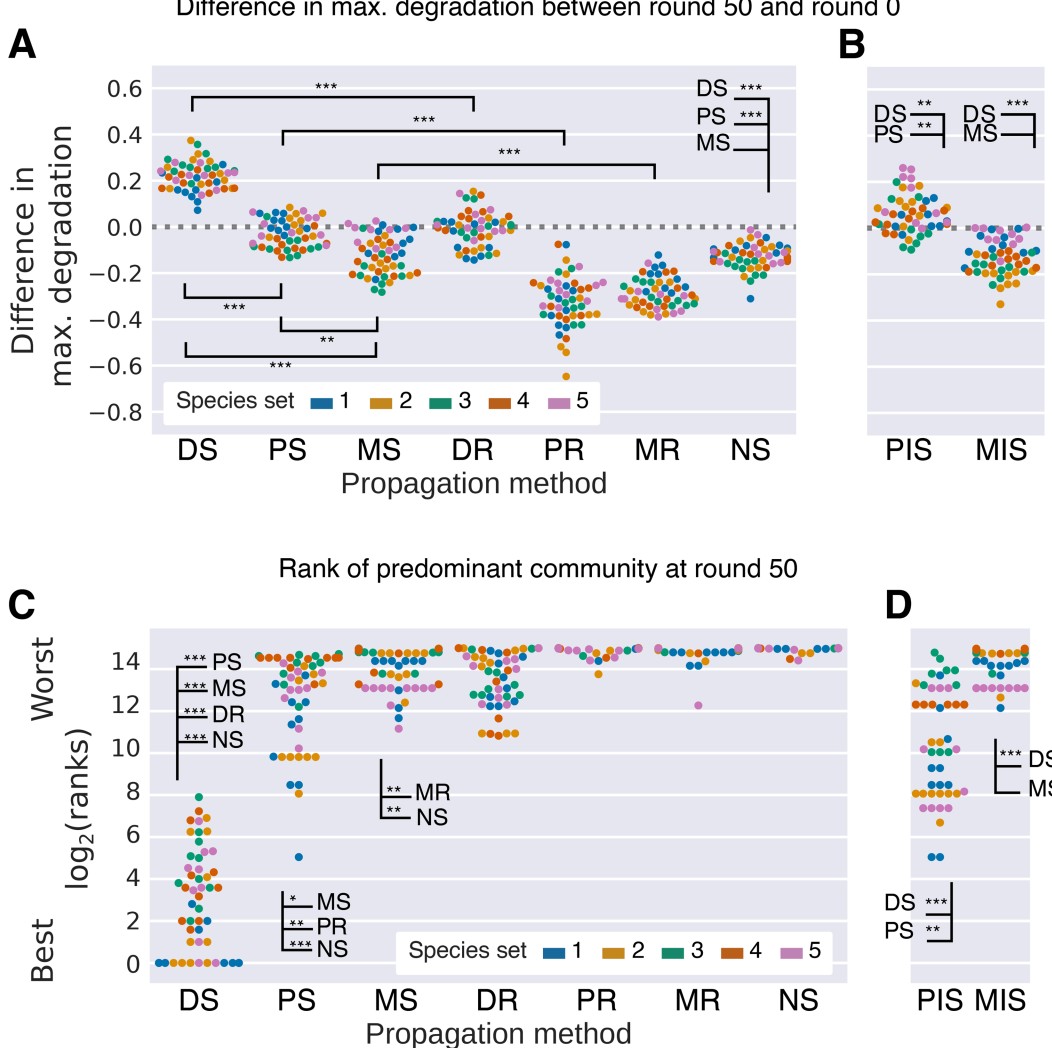

**Fig 2. Degradation scores and ranking of selected communities. (A, B)** The difference in maximum degradation score between round 50 and round 0 over the 21 communities. **(C, D)** The rank of the predominant community (the most common combination of species among the 21 communities in the last round of selection, not counting sub-communities) in terms of its degradation score compared to all of the 32767 possible combinations of 1, 2, ..., 15 ancestral species. **(B, D)** We reran the PS and MS propagation methods but where some communities receive invader species (PIS and MIS, respectively). This method was previously suggested by [26]. In all plots, each dot represents one of 10 repeated runs, colored by species set, with 50 dots in total. As each run starts from identical communities for all methods, we have compared pairs of runs between the selection methods. Asterisks show the significance of a Wilcoxon signed-rank test for difference in degradation between methods (*: $p < 10^{-3}$, **: $p < 10^{-6}$, ***: $p < 10^{-9}$). Corresponding plots generated using the ODE model are shown in S3 Fig and statistics are listed in S1 and S2 Tables.

migrant pool (MS) is in principle able to search all sub-communities of the first set of selected communities, they are in practice limited to a smaller subset as species tend to go extinct due to the toxic compounds, inter-species competition and/or the dilution bottleneck at each round. Accordingly, most communities available for selection by PS or MS resemble one another, seen as a rapid drop in between-community (or beta) diversity (Figs 4B and S8). In contrast, changing the species composition of some selected communities by inserting or removing species at random, DS can search a larger number of communities and the drop in beta diversity resulting from selection is not as steep. The beta diversity of the no-selection control depends on the diversity of the initial communities.

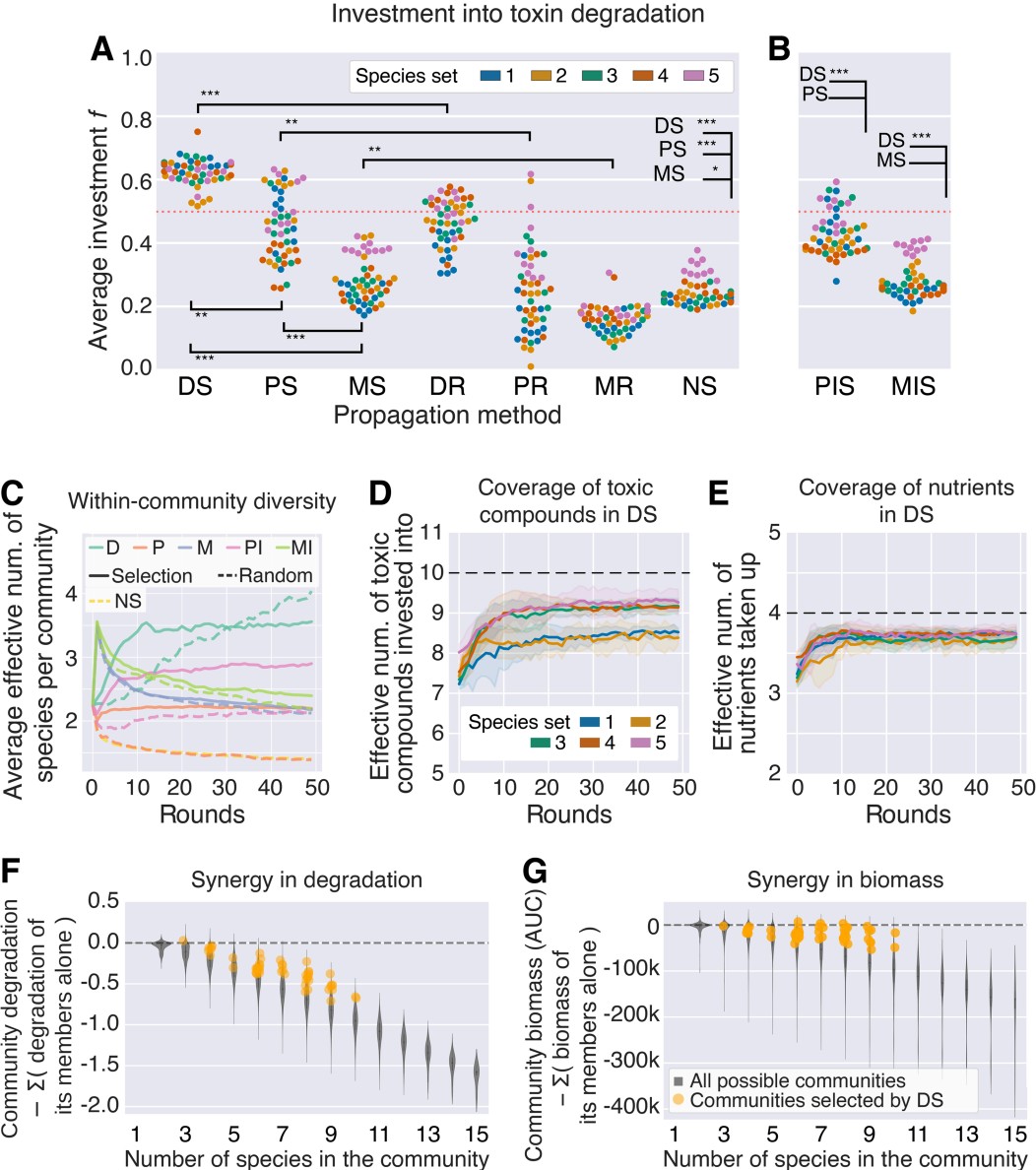

Fig 3. **Total investment into degradation, diversity, coverage and synergy. (A, B)** Average total investment in a community at round 50, averaged for the 21 communities in a run. For each species we calculate the average investment weighted by strain population and then we do the unweighted average of all the species in the community. The red dotted line at 0.5 indicates the theoretical mean investment at round 0. One dot for each of 10 repeated runs, colored by the 5 species sets, with 50 dots in total. The asterisks indicate the results of a Wilcoxon signed-rank test with $n = 50$ (*: $p < 10^{-3}$, **: $p < 10^{-6}$, ***: $p < 10^{-9}$). Corresponding plots generated using the ODE model are shown in S4 Fig and statistics are listed in S3 Table. **(C)** Within-community species diversity measured as the effective number of species (7), averaged over all 21 communities in a run. The line shows the average over the 10 repeats and the 5 sets of species, with error bars per set of species in S5 Fig. We also show changes in community richness in all methods in S6 Fig. **(D)** The coverage of toxic compounds and **(E)** nutrients in communities selected by DS, measured as the effective number invested into ($f_{ik}$) or taken up ($n_{ij}$) respectively within a community (mean ± s.d. over the 10 repeated runs for a given set of species), Table 1 (see Methods). **(F, G)** Synergy in communities selected by DS at round 50, grouped by community richness. Synergy is the difference in **(F)** degradation scores or **(G)** biomass between a co-culture and the sum of the values of the corresponding monocultures. The violin plot shows the distribution of synergy for each possible community of that richness level after one round of growth. The dots show the average synergy per repeated run in the last round for the 5 species sets selected by DS. The average species richness per repeated run is rounded to obtain an exact value. For visibility, we have plotted all species sets in the same color.

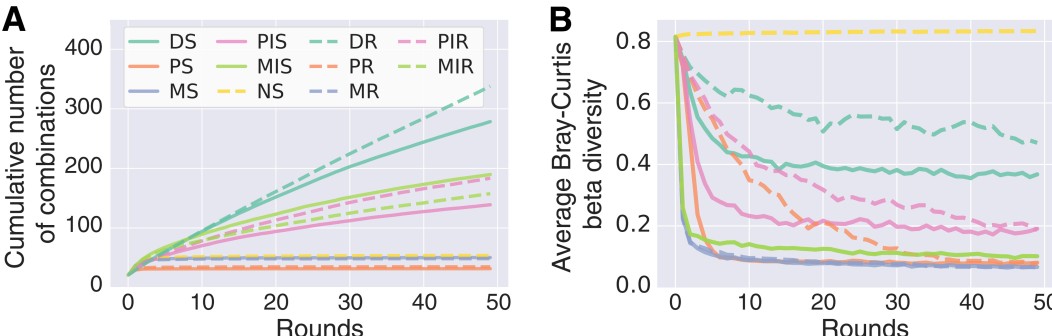

**Fig 4. Cumulative number of communities found by the selection methods and between-community diversity explain how DS can find better communities.** We show the mean over the 10 repeats and 5 species sets for each propagation method and refer to S7 and S8 Figs for the full results. Data shown here for the IBM model and in S9 Fig for the ODE model. **(A)** Time series of cumulative number of unique communities for each selection method. **(B)** Between-community or beta diversity, calculated as the average Bray–Curtis distance of each pair of communities.

## Propagule but not migrant pool selection performs better by periodically adding species to communities

In the disassembly method, more and better communities can be found by randomly adding and removing species in some of the communities. To explore whether species introduction could improve PS and MS in our models (previously shown for PS [26,28]), we implemented two new versions (PIS and MIS), where in each round, a fixed number of communities chosen at random will receive one or more "invader" species (also chosen at random) with a defined initial population size (see Methods).

With this modification, PIS increases the maximum degradation (one-sided Wilcoxon signed-rank test of $D$ in round 50 versus 0, $p < 10^{-3}$, $n = 50$, Fig 2B) and improves upon the standard PS method (two-sided Wilcoxon signed-rank test, $p < 10^{-6}$, $n = 50$). In our parallel implementation in the ODE model, however, the PIS method still improved upon the PS method (two-sided Wilcoxon signed-rank test, $p < 10^{-3}$, $n = 50$), but we did not find any significant improvements in the maximum degradation score compared to round 0 ($p = 0.9$, $n = 50$, S3 FigB). PIS finds higher-ranking communities than PS in both models (two-sided Wilcoxon signed-rank test for differences in ranks between PIS and PS, IBM: $p < 10^{-6}$, $n = 50$, Fig 2D, ODE model: S3 FigD) and can explore more combinations than the regular PS. Finally, the initial drop in beta diversity is less severe (Fig 4B), indicating that there is more variability for selection to act on. In contrast, MIS does not improve significantly on MS, either in terms of degradation, ranks or investment. Even though MIS explores more species combinations than MS, the beta diversity rapidly drops (Fig 4B), and the introduced species do not contribute much to diversity or degradation of the resulting communities.

## Mutation and selection can decrease per-species investment, but this increases biomass, maintaining community degradation

We have shown that DS can improve degradation by exploring many different species combinations and find ones that rank highly. Shuffling species around is, however, not the only way to improve degradation scores. Our models allow for mutations to the parameter $f$ that determines the trade-off between investment into degradation and biomass production for a cell. If a mutant is more competitive than its parent, it can replace the original type in future rounds, even as other species come and go around it. To investigate the effect of mutations, we compare the investment into degradation of species at round 50 to that of their ancestors from round 0, and analyze how these changes affect degradation at the community level.

In DS, the total per-species investment $\sum_k f_{ik}$ (abbreviated as $f_{i.}$) into degradation was significantly lower after 50 rounds of selection than that of the corresponding ancestral species (one-sided Wilcoxon signed-rank test of $f_{i.}$ in round

0 versus 50, $p < 10^{-6}$, $n = 50$, Figs 5A and S10). Given the trade-off between investing into growth versus degradation, the communities made up of evolved species had greater total biomass than communities composed of the corresponding ancestral species (one-sided Wilcoxon signed-rank test of total AUC in round 0 versus 50, $p < 10^{-9}$, $n = 50$, Fig 5B), such that overall, the degradation of the evolved communities was marginally but significantly higher (+$6 \times 10^{-3}$ units, averaged over all species sets, one-sided Wilcoxon signed-rank test $p < 10^{-3}$, $n = 50$) than that of communities made up of their ancestors (Figs 5C and S11). Compared to the improvement in degradation due to finding better species combinations, the improvement due to species evolution is very small and is not likely to have a large effect on the outcome of selection. To further quantify this effect, we reran our algorithm, setting mutation rate to 0. We observed no significant difference in the performance of our algorithm with and without mutation (comparing the difference in maximum degradation between generations 50 and 0 with mutation rate $\mu = 0.1$ and $\mu = 0$, one-sided Wilcoxon signed-rank test : $p = 0.19$ and $p = 0.64$ for DS and DR respectively, $n = 50$, S12 Fig).

In summary, the disassembly method improved the degradation scores over the 50 rounds of selection by finding better species combinations. Within those communities, individual species evolved to invest less into degradation and more into biomass production. As an effect of the trade-off between degradation and growth, the communities still maintain their degradation capabilities and the most efficient communities are the species combinations found in round 50, composed of either their ancestral or evolved genotypes. Whether or not species were allowed to evolve had no measurable effect on performance.

## Communities selected by DS are less stable than those selected by PS and MS

The disassembly method has features to ensure heritability and promote within-community diversity: we re-inoculate species in fixed and equal abundances, punish extinctions and re-inoculate extinct species. Controlling the ecological

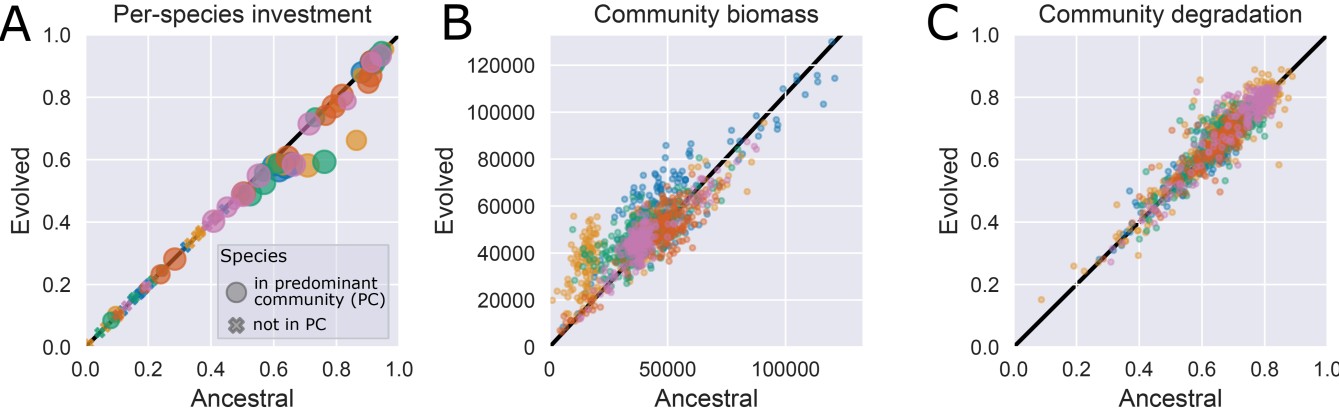

**Fig 5**. **Change in degradation investment per species and the effect of such change in the biomass and degradation of the communities in the last round, colored by species sets.** All results from the IBM. **(A)** Total investment $f_i = \sum_k f_{ik}$ for each species $i = 1, 2, ..., 15$. There are 75 dots: 15 species, colored by the species sets 1-5. For evolved condition, the markers show the average total investment of the species, weighted by population size for each occurrence in communities from the last round of selection where the species is present and not weighted between repeated runs; while ancestral condition corresponds to the total investment of each ancestral species prior to any growth or evolution. Species that were present in a predominant community (P.C., see definition in the caption of Fig 2) in the last round are shown as circles, where the radius is proportional to the number of repeated runs where the species appear in a P.C Species that were never present in a P.C. are represented by crosses. We summarize the p-values of Wilcoxon tests of whether the investment is different in the last round of selection compared to the first in S10 Fig. **(B)** Total biomass per community, measured as the sum of the area under the growth curves (AUC) for species in the community. The initial AUC is calculated from one round of growth, where the community is composed of ancestral strains of the same species in the same proportions as the last community. There are 1050 dots: 21 communities per 10 repeated runs, for each of the 5 species sets. **(C)** Degradation scores of the same 1050 communities in (B). The degradation of the initial community is calculated over one round of growth when the community is composed of ancestral strains of the same species.

PLOS Computational Biology

dynamics so tightly means that if we were to simply transfer these communities without adjusting relative abundances and without selection, as in the no-selection treatment, they could drift towards a different equilibrium with a lower degradation score. To assess the ecological stability of the selected communities, we transferred the best communities from round 50 for an additional 25 rounds of growth and dilution, this time without selection (Fig 6A) and found that the degradation scores of communities selected by DS dropped by $-0.21 \pm 0.14$ on average when left to their natural dynamics, close to how much the selection method increased the degradation ($0.22 \pm 0.06$). This indicates that the high performance of these communities relied on controlling the ecological dynamics. Once ecologically stable, the communities converge to a degradation score that is not significantly different to the average of the initial communities (one-sided Wilcoxon signed-rank test, $p = 0.24$, $n = 50$, Fig 2A, B).

In contrast, the degradation does not drop as much in communities selected by the classical methods PS and MS ($-0.02 \pm 0.03$ and $-0.03 \pm 0.03$ in max degradation, respectively, Fig 6A). The methods are stable in the sense that the communities do not change much after the first few rounds of selection, either in terms of composition (Fig 3E) or degradation (S13 Fig). The methods with invasion, PIS and MIS, show an intermediate drop in degradation ($-0.07 \pm 0.07$ and $-0.07 \pm 0.04$) indicating that the invasion step has an effect on community stability. In order to remain effective, the communities found by DS should be grown in the same conditions as they were selected, i.e. (i) from equal abundance, (ii)

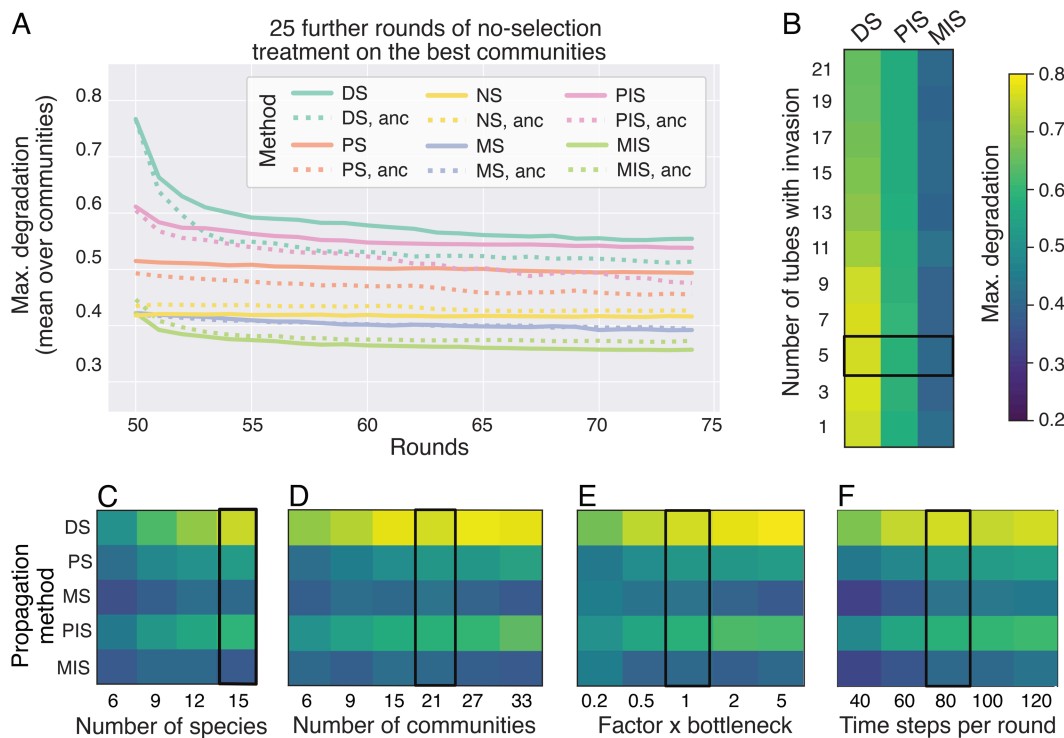

**Fig 6**. **(A) Stability in degradation over 25 additional rounds after releasing the selection pressure, calculated for the highest-scoring community for each selection method.** Results show the average degradation for each repeat. The dotted lines show the corresponding degradation when the community is composed of ancestral species. **(B-F)** Max. degradation score in the last round of selection as a function of the experimental parameters: **(B)** the number of tubes to receive or lose a species, **(C)** the number of species in the initial set, **(D)** the number of tubes or communities, **(E)** a scaling factor for the dilution bottleneck and **(F)** the number of time steps in each round. The heat maps show the median degradation score over all species sets and repeated runs (in (C) also over sub-samples) for each selection method. S14–S18 Figs show the full data set. The color bar is the same for panels B-F. The black outline marks the parameter value used throughout the rest of the paper. All data from the IBM.

without any intermediate rounds of growth in between rounds of selection. The latter has been suggested to stabilize the dynamics and improve community selection [26].

**Varying experimental parameters to decrease the size of the experiment**

Our models show that DS can outperform other propagation methods, as long as the ecological dynamics of the communities are controlled. However, DS is more cumbersome than the other methods from an experimental perspective: constantly dis- and re-assembling communities and having to adjust the population sizes of each species at every round can cost a lot of time and resources. We now investigate how four experimental parameters impact the degradation scores, and affect experiment size. We focus on DS but also compare it to the other methods (Figs 6B-F and S14–S18).

The parameter with the strongest effect on experiment size and the maximum degradation score is the number of species in the initial set (Spearman's rank correlation coefficient $\rho = 0.67$, $p < 10^{-9}$, S15 Fig). This means that the meta-community needs to be as rich as possible to efficiently improve degradation, and the main effort should be invested into managing a larger number of species, ideally by adding species that have positive effects on degradation or the growth of others. In contrast, the number of communities clearly affects experiment size, but it had a weaker correlation to degradation for DS ($\rho = 0.35$, $p < 10^{-9}$), meaning that the number of communities could be decreased, which would reduce effort with a limited effect on community performance.

The number of communities receiving an invading species is negatively correlated with degradation ($\rho = -0.29$, $p < 10^{-9}$). This is likely because too frequent invasions make it difficult to maintain high-performing species combinations. On the other hand, invasions cannot be reduced to 0, as DS relies on mixing species to explore the search space. The optimum appears then to lie in introducing species with a small, positive frequency (S14 Fig). Finally, the dilution factor (i.e. how large a fraction of the culture to re-inoculate for the next round of growth) is positively correlated to degradation scores ($\rho = 0.52$, $p < 10^{-9}$), and the number of timesteps per round appears to be close to the optimum at $t = 80$. When rounds are very short, cells do not have enough time to degrade, but if rounds become too long and the community is performing well, nutrients are likely depleted around $t = 80$ such that cells do not benefit from the additional time (S18 Fig).

## Discussion

The major challenges for community breeding are ensuring (i) that the community function is heritable, (ii) that within-community selection does not dominate over between-community selection and (iii) that communities differ sufficiently in phenotype to allow selection to act. While other theoretical studies have investigated heritability and the balance between within- and between-community selection [10,24–26], our "disassembly" method contributes to improving the third point: how to maintain variability between communities.

In our models, disassembly improved significantly upon the maximum degradation scores of simulated synthetic communities, compared to a random line and a no-selection control. The method further outperformed the classical propagule and migrant pool methods, which could only improve the maximum function for some initial combination of species, confirming previous findings [3,26]. Propagule selection has important advantages that we lose with disassembly selection: (i) resulting communities are ecologically stable and (ii) there is no need to culture individual isolates, allowing for obligate mutualists – a point we have not explored here. However, these methods suffer from the loss of between-community variability, which can be circumvented by periodically invading them with new species, as proposed previously [26,28] and confirmed here. The maintenance of diversity in the disassembly method is achieved by exchanging species between communities, an idea inspired by the crossover operator in genetic algorithms in the field of evolutionary computing [27, 33,34], whereby building blocks are recombined between digital individuals and generate variability for selection to act on.

Of course, crossover in genetic algorithms is itself inspired by recombination in sexual organisms [35,36]. While interesting, these parallels do not map directly to species exchange in community breeding, where the units that are subject to exchange are well-defined and have their own ecology and within-species evolution.

Our approach was also designed to address the second challenge of community breeding: reducing within-community and within-species selection compared to between-community selection. Beyond the practical goal of improving community performance, this question is also a fundamental one: can one apply selection such that group-level selection dominates over individual-level selection? We designed our algorithm for high group-level selection in two ways: first by competing communities against one another in each selection round and second by penalizing between-species competition by reducing the score of communities in which species were driven extinct. These decisions favored communities with more synergistic effects (Fig 3F and 3G). To follow how selection within species affects the outcome, we allowed species to acquire mutations in their investment into degradation (unlike other theoretical studies [26]). We found that competition between individuals is indeed strong in our simulated communities, leading species to evolve reduced investment into degradation in favor of growth. Interestingly, though, the faster growth rate of these evolved species compensated for the reduced investment by producing larger population sizes of degraders, such that the difference in degradation between ancestral and evolved communities was negligible. We expect this outcome to apply whenever community function is coupled to growth. Whether such a trade-off exists in experimental systems is therefore an important consideration to take into account.

The problem that we have not yet addressed then is heritability, which is crucial for evolution [37] and a major challenge for community selection [10]. As in [10], we sidestep the issue of heritability and ecological stability in the disassembly method by always re-assembling communities in a fixed abundance and equal species proportions. This allows community dynamics to unfold in almost the same way in each round during selection, such that offspring communities resemble their parents. But once the winning communities were found and propagated by regular growth and dilution, they were not ecologically stable and changed in both composition and function. One way to overcome this would be to include a few rounds of transferring between the rounds of selection to allow the communities to equilibrate before each selection step [26]. This would, however, make disassembly quite inefficient, in which case, propagule with invasion can be more stable than disassembly (Fig 6A). Another solution could be to restrict our search space towards communities that have higher heritability, i.e. communities that behave more like "evolutionary individuals" [38]. If we could accurately predict which species combinations are ecologically stable based on species' properties, we could restrict the algorithm to only considering those combinations. We would thereby aim to achieve individuality first, and selection second, increasing the likelihood that the winning community would be one that does not lose its properties after the end of the selection experiment [38].

Ultimately, the goal of our investigation was to help design community selection experiments. To explore all the factors that might matter in such experiments, we opted here to build models that are rather complex and strive to overcome this complexity by building two different models (IBMs and ODEs) to support the generality of our conclusions. The decision to allow for genetic mutations, for example, may seem excessive, but what we learned from it is important: in experimental systems where a trade-off between function and growth exists, within-species evolution may play a minor role in affecting performance. This means that experimentally, one need only work with ancestral species, avoiding the need to disassemble communities into their individual members. At each round then we would select the best communities, and rebuild them from the *ancestral* freezer collection after changing their composition with a given probability by removing or introducing species. This conclusion is also supported by our experimental work [31]. We also found that the parameter with the biggest effect on the method is the size of the initial species set and that the disassembly method is more efficient for larger sets of species (Fig 6C). If disassembling communities is not necessary, increasing the species pool would be much more feasible. Finally, we find that for disassembly, a higher number of communities grown in each selection round can compensate for the number of selection rounds needed since we are then able to search more distinct communities per round of selection.

We have made a number of simplifying assumptions in our models. First, we assume a well-mixed liquid culture, where in reality, clumps may affect species interactions and community function. We also assumed a trade-off between degradation and growth in both models and that the toxic compounds cannot be used as nutrients. Both assumptions serve to decouple the community phenotype from population growth and while they are not completely independent, a smaller contribution from growth on the community phenotype should make artificial selection more difficult. In a sense, this is the more interesting problem to explore, since all the methods we explored are expected to improve a community phenotype that is aligned with population growth. Further, we have assumed that species diversity is key to functional diversity: each species can only degrade a subset of the toxic compounds in our model and complete removal of the compounds depends on finding other species with complementary degradation capabilities. Within-community diversity is in this way fundamental for community success, and also decreases competition, as each species only uses a subset of the nutrients. In our simulations, it is therefore unlikely that any mono-culture scores higher than a multi-species community, and the median size of a selected community was $6.9 \pm 1.8$ species after 50 rounds of selection by disassembly (in the IBM model, S6 Fig). This optimum will, however, differ for each system [30].

Taken together, we have computationally explored an approach to community selection, where species composition is shuffled between competing communities, allowing for a greater exploration of the space of possible communities to find the best performing ones. In doing so, we have shown that community function can improve with respect to randomly assembled communities, but that genetic mutation can contribute to reduced investment by individual strains into community function.

## Materials and methods

We separately implemented an individual-based model (IBM) of individual cell growth and a system of ordinary differential equations (ODEs) that model population-level dynamics. We chose to describe the IBM in the main text and the ODE model in S1 Note, as the ODE model obliged us to fix an upper limit to the number of strains in a community (number of differential equations), making it difficult to implement mutations and the migrant pool method. The ODE model therefore only recapitulates a subset of the results. All parameters are listed in Tables 1 and 2. All code related to this manuscript can be found at https://github.com/Mitri-lab/artif_comm_select_model.git.

### Setting up the simulations

A set or pool of 15 species (the "metacommunity") is generated prior to the start of the simulations by fixing parameters for each species $i$ describing its intrinsic growth rate ($a_i$, $r_i$), its preference for each nutrient $j$ ($n_{ij}$), how it is affected by ($m_{ik}$) and degrades ($f_{ik}$) each toxic compound $k$ (S2 Algorithm, Table 1, and Fig 1A).

We then prepare 21 simulated well-mixed batch culture tubes. At $t = 0$ in the very first round, each culture tube contains $S_i = 10$ identical cells of each of 4 microbial species, $T_k(t_0) = 700$ units of each of the 10 toxic compounds, which can cause cell death, and $N_j(t_0) = 2000$ units of each of the 4 nutrients that allow the cells to divide and reduce the concentrations of toxic compounds (Fig 1A). We assign the 4 species in each tube by selecting randomly with replacement from the pool of 15 species, making sure that each species is present in at least one of the 21 communities. In some simulations, the number of species can vary. When this happens, we inoculate 10 cells of each species thereby changing the total inoculum. We verified that this choice (as opposed to keeping the total constant and partitioning the population among the different species) did not have a strong effect on performance (S19 Fig).

### Toxic compound degradation, cell division, nutrient consumption, and death

Each round of selection consists of 80 simulated time-steps. At each time-step, growth, cell division, and death occur according to probabilities that are fixed for each species $i$ when the initial species set is generated (Table 1), and these probabilities are used for random sampling from a Poisson distribution [25]. How much a cell of type $i$ invests into toxic

**Table 2.** Other parameters used in the paper, and relevance to the different propagation methods. DS = Disassembly Selection, DR = Disassembly Random, PS = Propagule Selection, PR = Propagule Random, PIS = Propagule Selection with Invasion, PIR = Propagule Random with Invasion, MS = Migrant pool Selection, MR = Migrant pool Random, MIS = Migrant pool Selection with Invasion, MIR = Migrant pool Random with Invasion, NS = No Selection. We explore variations to some of these in Fig 6.

| Description | Value | Relevant propagation method |
|---|---|---|
| Mutation rate $\mu$ | 0.01 | all |
| Mutation magnitude | lognormal [0, 0.4] | all |
| Number of tubes | 21 | all |
| Selected parent tubes in each round | 7 | all except NS |
| Size of species pool | 15 | all |
| Number of nutrients | 4 | all |
| Initial nutrient concentrations [$N_j(t = 0)$] | 2000 | all |
| Number of toxic compounds | 10 | all |
| Initial toxic compound concentrations [$T_k(t = 0)$] | 700 | all |
| Toxic effect Hill function coefficient $K$ (Equation 3) | 700 | all |
| Initial number of cells (round 0) [$S_i = p_{i0} = 10$] | 10 | all |
| Initial number of cells (all rounds) [$S_i = p_{i0} = 10$] | 10 | DS, DR |
| Bottleneck size (number of cells transferred) | 1/20 of prev. round | PS, PIS, PR, PIR |
| Bottleneck size (number of cells transferred) | 1/21 of prev. round, pooled | MS, MIS, MR, MIR |
| Time steps $t$ per round ($t_{end}$) | 80 | all |
| Number of selection rounds | 50 | all |
| Number of replicate simulations per initial condition | 10 | all |
| Number of tubes to receive an invader | 5 | PIS, MIS, DS, DR |
| Number of tubes to lose a member | 5 | DS, DR |

compound degradation (from the nutrients it consumes) is determined by $0 \leq \sum_k f_{ik} \leq 1$ (we abbreviate $\sum_k f_{ik}$ as $f_{i\cdot}$). These $f_{ik}$ parameters can change if a cell mutates, creating new lineages $i$ of the same ancestral species that are identical in all other parameters. The remaining fraction of nutrients a cell takes up $1 - f_{i\cdot}$ is invested into population growth, creating a trade-off between growth and toxin degradation.

At each time-step, toxic compound degradation occurs first, followed by cell growth, then death. Cell growth consists of two steps, "activation" and "replication" [39]. At a given time-point $t$, a culture contains a population of size $p_{i0}$ of inactivated cells, and $p_{i1}$ of activated cells of each lineage $i$ ($S_i = p_{i0} + p_{i1}$). At $t = 0$, all cells are inactive, and every time a cell divides, its daughter cells are both inactive. Only degradation and activation require nutrient consumption (i.e. they are costly).

We first calculate how much nutrient will be consumed by each species for degradation and activation, which depends on the preferences $n_{ij}$ of species $i$ ($\sum_j n_{ij} = 1$), and the availability $N_j$ of each nutrient $j$. If for at least one nutrient, $N_j < n_{ij}$, we set that $n_{ij} = 0$ for the rest of the round, compute the maximum amount of energy that the species can take up at that time-step $u_i = \sum_j n_{ij}$ and re-scale the preferences of that species to $\hat{n}_{ij}$:

$$\hat{n}_{ij} = \frac{n_{ij}}{\sum_j (n_{ij} \text{ if } N_j > n_{ij})}, \tag{1}$$

such that $\sum_j \hat{n}_{ij} = 1$. This implementation ensures that even if some nutrient types get depleted, cells consume the remaining required nutrients, even if less efficiently because $u_i < 1$. Next, we calculate the maximum number of cells of that species that can afford to consume nutrients, based on current availability: $S_i^{max} = \min_j(N_j/\hat{n}_{ij})$.

Toxic compound degradation occurs first. For each toxin $k$, $S_i^{max}$ cells of each lineage $i$ (or $S_i$ if it is $< S_i^{max}$) degrade the amount $f_{ik} \cdot u_i$ units, consuming $\hat{n}_{ij} \cdot f_{i.}$ units of nutrients. When a toxic compound is depleted, its degradation and the corresponding nutrient consumption do not occur.

The next step is cell activation. An inactive cell can activate with probability

$$a_i(1 - f_{i.})u_i \sum_j \left( \hat{n}_{ij} \frac{N_j(t)}{N_j(t_0)} \right), \tag{2}$$

where $a_i$ is the fixed activation probability for a given species. This expression is used to sample from $S_i^{max}$ (or $p_{i0}$ if it is $< S_i^{max}$), resulting in the number of cells that activate, thereby consuming $\hat{n}_{ij}(1 - f_{i.})$ units of each nutrient, and joining the active population $p_{i1}$. All cells that activated in previous time-steps but did not replicate continue to consume the same amount of nutrients to remain activated.

Next, activated cells $p_{i1}$ divide with probability $r_i(1 - f_{i.})$ without additional nutrient consumption, each yielding two inactivated daughter cells: one daughter maintains the parameter values, and the other is susceptible to mutation with probability $\mu = 0.01$ to mutate each of its $f_{ik}$ values. Upon mutation, the previous value of $f_{ik}$ is multiplied by a random number from the lognormal (mean$= 0, \sigma^2 = 0.4$) distribution, making sure the total investment $f_{i.}$ falls in the $[0, 1]$ interval. As a result, a new lineage of the same species with population size $p_{i0} = 1$ is introduced.

Activated and inactivated cells may die with a probability determined by the following Hill function:

$$\sum_k m_{ik} \frac{T_k^2}{T_k^2 + K^2} \tag{3}$$

Where $T_k$ is the current concentration of toxic compound $k$ and the constant $K = 700$.

## Degradation score

Each round consists of $t_{end} = 80$ time-steps, in which the cells in each culture tube grow and degrade the toxic compounds independently of the others. At the end of the round, we calculate the degradation score $D$ as the root-mean-square decrease in $T_k$ from the initial time point $t_0$ to the last $t_{end}$:

$$D = 1 - \sqrt{\frac{1}{10} \sum_{k=1}^{10} \left( \frac{T_k(t_{end})}{T_k(t_0)} \right)^2}. \tag{4}$$

These scores are used to rank the communities, so that we can select the best ones and use them to start a new round. The "offspring" culture tubes in the new round again contain $T_k(t_0) = 700$ of each toxic compound, $N_j(t_0) = 2000$ of each nutrient and are seeded with cells from the selected tubes from the previous round – the "parent" communities – according to the propagation method (see below). We simulate a total 50 rounds of growth, degradation and selection.

## Simulation replicates

The events occurring at each time-step are stochastic and lead to noise between runs of the model with the same starting conditions. The outcome also depends strongly on the parameters of the set of species that we sample from. We therefore generate 5 sets of 15 species each (we sample $5 \times 15 = 75$ times from Table 1). For each set, we then run 10 simulations, each starting with different combinations of the 15 species. The same exact initial conditions are used for all selection methods to make comparisons fair. When we calculate the growth and degradation of specific communities such as the 32767 possible combinations of species (Figs 2E-F and 3G), we average the results over three replicates of the simulations. We use the same seed for the random number generators throughout all simulations for consistency.

## Propagation methods

After scoring all communities in a given round (Equation 4), we select 7 out of 21 parent communities that are propagated to create offspring for the next round of growth (Fig 1A and  S1 Algorithm). Here we implement several methods of propagating the selected communities, and compare them to one another (Fig 1B). The choice of bottleneck size (7/21) is relatively wide, as compared to similar studies in the literature [9,10,12–17,19–22,25,26,40–42].

**No selection (NS).**  We implement the no-selection treatment as a control or a baseline, to evaluate how the community function (toxic compound degradation) changes simply due to interspecies interactions and the resulting ecological and evolutionary dynamics [26]. Note that all interspecies interactions are due to the consumption of nutrients and degradation of toxic compounds. NS is the only treatment where we do not select 7 of the parent communities. Instead, each community is propagated by approximately 20-fold dilution at the end of each round. Dilution occurs by stochastically sampling cells from the parent population with all cell types combined according to a Poisson distribution and placing them in the offspring tubes. The dilution factor is approximate as we have a discrete number of cells in each population, which is not always divisible by 20.

**Propagule selection (PS).**  In propagule selection, the 7 communities with the highest scores are propagated to the next round by dilution [13,29] (Fig 1B, S7 and S8 Algorithms). Each selected parent contributes equally to 3 offspring communities. We then seed each offspring by diluting the corresponding parent community by approximately 20, as in NS. This means that following the first round (where we inoculate 10 cells of each of 4 species), every tube may contain a different number of species, each with a different number of cells, and a different total population size. We compare PS to the random control PR, where 7 communities are selected at random, ignoring their degradation scores. We also compare PS to a version that we call propagule with invasion (PIS) and its corresponding random control (PIR). In this version, we introduce at least one species (chosen at random with uniform probability out the set of 15), to 5 out of the 21 offspring communities (chosen at random with uniform probability) [26].

**Migrant pool selection (MS).**  In MS, the 7 selected parent communities are mixed in a *migrant pool* before new offspring communities are formed by sampling from the pool [13,29] (Fig 1B, and  S9 Algorithm). To create the 21 offspring communities, cells are sampled without replacement from the pool with an approximately 20-fold dilution (stochastic process according to Poisson distribution). In MS, as with PS, the species composition and cell number can vary compared to the initial 10 cells of 4 species in each tube of the first round. In contrast to PS, though, all tubes in following rounds are expected to have similar total population sizes, as they are all seeded from the same pool.

We compare MS to a random control (MR), where we select the 7 communities with uniform probability ignoring their degradation scores. We also implement a version of migrant pool selection where we introduce one or more species (chosen at random with uniform probability out the set of 15) to 5 out of the 21 offspring communities (chosen at random with uniform probability), and call this migrant pool with invasion (MIS and the random control MIR).

**Disassembly selection (DS).**  We have designed this selection method to overcome some of the short-comings of the other selection methods (red box in Fig 1B and  S10 Algorithm). Rather than diluting each parent to create an offspring, DS disassembles each parent community into its member species (including each species' different lineages) and uses

these isolated species to recompose communities that resemble the parents in species composition, with some variability introduced by removing species that were present in the parent or introducing new member species from the species pool. By disassembling communities, we maintain a record of samples of each species and their evolved lineages that were present in at least one selected community in each round (box outlined in grey in Fig 1B, DS). If a species is present in more than one selected community, we sample from the highest-scoring community that this species was part of.

Another difference with DS is that we modify the scoring method by penalizing communities in which extinctions occur. The rationale is to favor communities whose member species do not out-compete one another, which we expect to lead to more cooperative communities, potentially with better collective functions. We implement this by scaling the degradation score $D$ by the fraction of surviving species at the end of a round of growth as follows:

$$\hat{D} = D \times \frac{\text{number of surviving species}}{\text{number of species in the community}}.$$ (5)

For example, if a 5-species community loses one member species, its degradation score is scaled by 0.8. Next, we draw 21 offspring communities from the 7 ($n = 1, ..., 7$) selected parent communities for the next round of growth, in proportion to their scores $\hat{D}$ with probability:

$$\frac{1}{\sum_k \hat{D}_k} \hat{D}_n,$$ (6)

with replacement.

We then modify the species composition of the 21 new offspring communities to introduce variability between communities. First, we choose 5 of the 21 offspring communities at random and remove one or more species, always one species plus an additional number drawn from Poisson(0.5). This a choice which means that we are more likely to remove few species but can also remove more from time to time. If the drawn number is equal to or higher than the number of species currently in the community, we leave one species to avoid emptying or completely changing the community composition. Having found a number of species to remove, we choose the species to remove with uniform probability, but avoid removing any species that is present in only this community. Next, we randomly chose 5 communities to which we introduce one or more invader species —as above, 1 + Poisson(0.5)— chosen with uniform probability from the species pool. These 5 are chosen anew and could be the same communities that we just removed species from, or not. In order to maintain diversity, we ensure that all species appear in at least one community by preferentially introducing species that are not currently present in any offspring community. If a species goes extinct in all 21 tubes, we re-introduce it from the most recent round in which it was present. See S10 Algorithm for more details.

Once the final species composition of each offspring community has been determined, 10 cells of each species from the species record (box outlined in grey in Fig 1B, DS) (sampled randomly with replacement from a Poisson distribution) are inoculated into the offspring tubes. The initial population size in all tubes is therefore always 10 times the number of species, which begins with 4 in round 0, but can vary over subsequent rounds. Keeping the initial population size of each species fixed allows to standardize the growth conditions between rounds, i.e. to maintain heritability [10].

## Statistical and other analyses

Correlations are evaluated by the Spearman's rank correlation coefficient $\rho$. We compare selected communities to the set of all possible communities by a Kruskal–Wallis $H$ test for differences in median. We use the scipy [43] implementations for both.

We quantify species diversity within a community (Figs 3C and S5) as the Hill number of order 1 or the average effective number of species present in the community [44,45], which is based on the Shannon index $H'$:

$$\exp(H') = \exp\left(-\sum_i p_i \log p_i\right),$$ (7)

which in turn depends on the species' relative abundances

$$p_i = S_i/S_{tot}$$ (8)

where we divide the population size $S_i$ of each species $i$ by the total population size in the community $S_{tot} = \sum_i S_i$. If more than one lineage is present, we sum up their population sizes to find the species' total population size. We then average $\exp(H')$ over all communities to find the average effective number of species. The measure falls between 0 and 15 effective species in an average community.

Beta diversity (Figs 4B and S8) is calculated by considering each community as a vector of the population sizes of the 15 species. Species absence from the community is marked by zero. We find the beta diversity as the average Bray-Curtis dissimilarity of each of the 210 possible pairs in the 21 communities.

The community coverage of nutrients and toxic compounds (Fig 3D-E) is quantified similarly to species diversity, as the effective number of toxic compounds invested into or nutrients taken up. Toxic compound coverage is calculated from the vector $\tilde{f}_{.k} = \sum_i \tilde{f}_{ik}$ of total investment into degrading toxic compound $k$ in a given community. The 'tilde' indicates that we have scaled the $f_{ik}$ for each lineage of a species by the corresponding population sizes of the lineage, as in (10). In this way, we emphasize the most relevant lineage of each species and do not bias the result to the number of competing lineages within a species. Note that we do not scale $\tilde{f}_{.k}$ by species abundance in the community. Once each investment $\tilde{f}_{.k}$ is rescaled so that $\sum_k \tilde{f}_{.k} = 1$, we calculate the effective number of toxic compounds invested into as

$$\exp(H') = \exp\left(-\sum_k \tilde{f}_{.k} \log \tilde{f}_{.k}\right),$$ (9)

and average this value over all 21 communities. The measure falls between 0 (no toxic compounds are invested into) and 10 (all compounds). For the nutrient coverage, we use the same calculation using the nutrient uptake rates $n_{ij}$, but without scaling between different lineages as this parameter does not mutate. The effective number of nutrients taken up takes values between 0 and 4, the number of different nutrients.

To evaluate the evolution of the total investment $f_{i.} = \sum_k f_{ik}$ (Fig 5A) of species $i$, we compare the investment of the ancestral lineage to that in the last round of selection (or as late as possible in case the species went extinct). The average per-species investment is weighted by the population size $S_i$ of the different lineages $l$ of species $i$,

$$\tilde{f}_{i.} = \sum_{l \text{ of species } i} S_l f_{l.} / \sum_l S_l$$ (10)

to emphasize the investment of abundant lineages instead of small recent mutants that have not contributed as much to the community function. The value $\tilde{f}_{i.}$ is further averaged over the 10 repeated runs (recall: each run starts from the same set of species) for each of the 15 species, without weighting by population size. For the statistical comparison, we hence have 75 data points: 15 species per each of the 5 sets.

To evaluate the stability of the selection methods (Fig 6A), we choose the highest-scoring community that each method found after 50 rounds of selection (one community for each repeated run of each species set), and seed 10 replicates with

its identical initial composition. Then we grow and dilute them for a further 25 rounds, as we would do for the no-selection treatment. We do not allow mutations in these rounds, to focus only on the ecological stability of the found communities.

For the analysis of sensitivity to the number of species in the initial species pool (Figs 6C and S15), we sample 5 random subsets of 6, 9, 12 from the original set of 15 species, and run for each of them 5 simulations with different 21 initial communities. Drawing new species sets of the corresponding size would introduce further variance, which we would rather avoid. For the effect of the dilution factor (Figs 6E and S17), we multiply the 10-cell inoculum by a factor 0.2, 0.5, 1, 2 or 5 for the disassembly method and scale the 5% dilution fraction for the other methods by the same factor.

## Supporting information

**S1 Table** P-values for Figs 2A-B and S3A-B a Wilcoxon signed-rank test of difference in maximum degradation between methods.
(XLSX)

**S2 Table** P-values for Figs 2C-D and S3C-D a Wilcoxon signed-rank test of difference in degradation ranks between methods.
(XLSX)

**S3 Table** P-values from a Wilcoxon signed-rank test of difference in total investment in communities, between methods for Figs 3A-B and S4A-B.
(XLSX)

**S4 Table** Parameters defining a microbial strain $i$ in the ODE model. Growth and degradation parameters in relation to nutrients and toxic compounds $N_j$ and $T_k$. All parameters are assumed to be positive, and the investment $f_{ik}$ is limited to the interval [0, 1]. The matrices of growth rates, death rates and degradation investment $r_{ij}$, $m_{ik}$ and $f_{ik}$ are made sparse by multiplying them by matrices drawn from Bernoulli(0.5), i.e. flipping a coin for each entry. In this way, each species takes up approximately half of the nutrients, is affected by half of the toxic compounds and degrades half of the toxic compounds.
(PDF)

**S1 Fig** Illustration of the variables and processes in the ODE model. Populations of cells vary in their preferences for nutrients and degradation capabilities. The populations use the available nutrients to degrade the toxic compounds and to grow.
(PDF)

**S2 Fig** The difference in **median** degradation between round 50 and round 0 for each propagation method, corresponding to Fig 2 where the difference in maximum degradation is shown. The two-sided Wilcoxon test for difference in degradation against the no-selection control is significant for the selection methods DS, PS and MS. Data generated by the IBM.
(PDF)

**S3 Fig** Degradation scores and ranking of selected communities. (A, B) The difference in maximum degradation between round 50 and round 0 for each propagation method, corresponding to Fig 2, but generated by the ODE. (C, D) The rank of the predominant community (the most common combination of species among the 21 communities in the last round of selection, not counting sub-communities) in terms of its degradation score compared to all of the 32767 possible combinations of 1, 2, ..., 15 ancestral species, corresponding to Fig 2, but generated by the ODE.
(PDF)

**S4 Fig** Average total investment into degradation of communities at round 50, averaged for the 21 communities in a run, corresponding to panel A, B in Fig 3, but generated by the ODE.
(PDF)

**S5 Fig** Time-series of the species diversity (effective number of species per community) corresponding to Fig 3C. Each panel shows the mean $\pm$ standard deviation over the 10 repeated runs, for each species set 1-5, for one propagation method. Data generated by the IBM.
(PDF)

**S6 Fig** Time-series of the mean number of species per community (richness) corresponding to Fig 3C. This includes all communities in a round. Each panel shows the mean $\pm$ standard deviation over the 10 repeated runs, for each species set 1-5, for one propagation method. Data generated by the IBM.
(PDF)

**S7 Fig** Time-series of the number of explored communities, corresponding to Fig 4A. Each panel shows the mean $\pm$ standard deviation over the 10 repeated runs, for each species set 1-5, for one propagation method. Data generated by the IBM.
(PDF)

**S8 Fig** Time-series of the beta diversity corresponding to Fig 4B. Each panel shows the mean $\pm$ standard deviation over the 10 repeated runs, for each species set 1-5, for one propagation method. Data generated by the IBM.
(PDF)

**S9 Fig** (A) Time-series of cumulative number of unique communities for each selection method, corresponding to Fig 4A but generated using the ODE model. Time-series of the beta diversity corresponding to Fig 4B, but generated using the ODE model. Each panel shows the mean $\pm$ standard deviation over the 10 repeated runs, for each species set 1-5, for one propagation method.
(PDF)

**S10 Fig** Distribution of p-values from a one-sided Wilcoxon signed-rank test of whether the total investment $f_i$ of a species is larger/smaller in the last round where a species survived, than the investment of the ancestral species. There is one bar for each selection method, with 15 species x 5 sets of species for each bar. The alternative hypothesis is that difference in investment (ancestral-evolved) is greater (green) or less (blue) than zero. Data generated by the IBM. Data for DS is shown in Fig 5.
(PDF)

**S11 Fig** Histogram of difference in max degradation between evolved and ancestral communities. Triangles indicate the mean values for each species set. Data generated by the IBM.
(PDF)

**S12 Fig** The difference in maximum degradation between round 50 and round 0 for DS and DR, corresponding to Fig 2, but varying the mutation rate from the default of $\mu = 0.1$ to $\mu = 0$. The performance of the method was not significantly affected by this change in mutation rate.
(PDF)

**S13 Fig** Time-series of max. degradation over 50 rounds of selection, for the different propagation methods. Each plot corresponds to one species set and shows the maximum degradation in the meta-community averaged over repeats, with the standard deviation in shades of the corresponding color. For each species set, each repeat etc, the degradation score

at transfer 50 forms the swarms in Fig 2. The black line shows the degradation score of the best ancestral community out of the 32767 combinations. Data generated by the IBM.
(PDF)

**S14 Fig** Effect on max community degradation score from changing the number of communities to receive a migrating species. Data generated by the IBM.
(PDF)

**S15 Fig** Effect on max community degradation score from changing the number of species in the ancestral community. Different marker shape indicates sub-sample (1-5) for each species group of size 6, 9, 12. For 15 species we keep the original species sets. Data generated by the IBM.
(PDF)

**S16 Fig** Effect on max community degradation score from changing the number of communities. Data generated by the IBM.
(PDF)

**S17 Fig** Effect on max community degradation score from scaling the dilution factor or inoculum size. Data generated by the IBM.
(PDF)

**S18 Fig** Effect on max community degradation score from changing the number of time steps in each round. Data generated by the IBM.
(PDF)

**S19 Fig** The effect of changing the inoculum size on community performance. Left: Jaccard distances between the winning communities in each species set after running 32767 species combinations with either 10 cells per species or 150 cells total. Dark grey '1's and light grey '0's indicate presence or absence of each of the 15 species in the winning community. This shows that the two ways of initializing our tubes resulted in very similar winning communities. Right: Number of species in each of the winning communities colored by species set in the two seeding conditions. Note that the data in the left boxplot correspond to the yellow points in Fig 3F. Data generated by the IBM.
(PDF)

**S1 Algorithm** Overall flow of the selection simulations from population growth, community dynamics and mutations to propagation by the different selection methods.
(PDF)

**S2 Algorithm** Implementation of population growth, competition and mutations in the IBM model described in the section *Individual-based model*.
(PDF)

**S3 Algorithm** Activation of cells, the first step of cell division in the IBM described in S2 Algorithm.
(PDF)

**S4 Algorithm** Replication and mutation, second step of cell division of the IBM described in S2 Algorithm.
(PDF)

**S5 Algorithm** Cell death of the IBM described in S2 Algorithm.
(PDF)

**S6 Algorithm** Implementation of population growth, competition and mutations in the ODE model described in the section *Population-level model*. To solve the equations, we use *dopri5* from the SciPy library [43,46].
(PDF)

**S7 Algorithm** Implementation of the propagule selection method for the ODE model.
(PDF)

**S8 Algorithm** Implementation of the propagule selection method for the IBM.
(PDF)

**S9 Algorithm** Implementation of the migrant pool selection method for the IBM.
(PDF)

**S10 Algorithm** Implementation of the disassembly method.
(PDF)

**S11 Algorithm** Method to propose new communities based on their degradation scores from the previous round. Called by  S10 Algorithm.
(PDF)

**S1 Note** Supplementary methods on the implementation of the ODE model.
(PDF)

## Acknowledgments

We thank the Mitri lab at the University of Lausanne for valuable discussions, in particular Afra Salazar de Dios and Shota Shibasaki for detailed comments on the manuscript.

## Author contributions

**Conceptualization:** Björn Vessman, Flor Inés Arias-Sánchez, Sara Mitri.

**Funding acquisition:** Sara Mitri.

**Investigation:** Björn Vessman, Pablo Guridi-Fernández, Sara Mitri.

**Methodology:** Björn Vessman, Pablo Guridi-Fernández.

**Software:** Björn Vessman, Pablo Guridi-Fernández.

**Supervision:** Sara Mitri.

**Validation:** Sara Mitri.

**Visualization:** Björn Vessman, Pablo Guridi-Fernández.

**Writing – original draft:** Björn Vessman, Pablo Guridi-Fernández, Sara Mitri.

**Writing – review & editing:** Björn Vessman, Pablo Guridi-Fernández, Sara Mitri.

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
