## [Decision Letter · Decision Letter 0]

22 Jul 2025

PCOMPBIOL-D-25-01028

Novel artificial selection method improves function of simulated microbial communities

PLOS Computational Biology

Dear Dr. Mitri,

Thank you for submitting your manuscript to PLOS Computational Biology. After careful consideration, we feel that it has merit but does not fully meet PLOS Computational Biology's publication criteria as it currently stands. Therefore, we invite you to submit a revised version of the manuscript that addresses the points raised during the review process.

Please submit your revised manuscript within 60 days Sep 21 2025 11:59PM. If you will need more time than this to complete your revisions, please reply to this message or contact the journal office at ploscompbiol@plos.org. Please include the following items when submitting your revised manuscript:

We look forward to receiving your revised manuscript.

Kind regards,

Chaitanya S Gokhale, PhD

Academic Editor

PLOS Computational Biology

Tobias Bollenbach

Section Editor

PLOS Computational Biology

**Additional Editor Comments:**

Dear Authors,

Thanks a lot for submitting your manuscript to PLoS CB and waiting patiently for the reviewers. I was expecting ot get three reviewers, but on the quality of the two reviews that we have received so far, and not waiting longer for the third, here we go.

The reviewers have done quite a thorough job in my opinion and reflect some of the concerns that I had about the manuscript as well.

For example, the comment of Reviewer 1 on checking the performance differences according to the timescales of selection, or the comment of Reviewer 2 on the null case without migration.

I would recommend going point by point over the reviewer's comments, and while some may be easier to answer than others, I perceive this to be a major revision that may take some time, but it will definitely be a great read!

Looking forward to your next draft.

Best,

Chaitanya

**Journal Requirements:**

At this stage, the following Authors/Authors require contributions: Björn Vessman, Pablo Guridi-Fernández, Flor Inés Arias-Sánchez, and Sara Mitri. Please ensure that the full contributions of each author are acknowledged in the "Add/Edit/Remove Authors" section of our submission form.

5) We notice that your supplementary Figures are included in the manuscript file. Please remove them and upload them with the file type 'Supporting Information'. Please ensure that each Supporting Information file has a legend listed in the manuscript after the references list.

1) Please clarify all sources of financial support for your study. List the grants, grant numbers, and organizations that funded your study, including funding received from your institution. Please note that suppliers of material support, including research materials, should be recognized in the Acknowledgements section rather than in the Financial Disclosure

2) State the initials, alongside each funding source, of each author to receive each grant. For example: "This work was supported by the National Institutes of Health (####### to AM; ###### to CJ) and the National Science Foundation (###### to AM)."

3) State what role the funders took in the study. If the funders had no role in your study, please state: "The funders had no role in study design, data collection and analysis, decision to publish, or preparation of the manuscript."

4) If any authors received a salary from any of your funders, please state which authors and which funders..

7) Please ensure that the funders and grant numbers match between the Financial Disclosure field and the Funding Information tab in your submission form. Note that the funders must be provided in the same order in both places as well.

**Reviewers' comments:**

Reviewer's Responses to Questions

**Comments to the Authors:**

Reviewer #1: This manuscript proposes a new selection method to improve the functional performance of microbial communities through artificial selection. The key challenges in this area involve ensuring group-level trait inheritance and maintaining sufficient between-group variation. The proposed method, called "disassembly selection (DS)," addresses these challenges by actively increasing between-group variation—achieved through the addition or removal of species in offspring communities. This strategy enables the exploration of a wide range of species combinations, thereby enhancing the likelihood of discovering high-functioning community compositions. The manuscript demonstrates that this method outperforms conventional selection strategies. I enjoyed reading the manuscript and found that the results are both interesting and impactful for the field of microbial community evolution. However, the dynamics of the model system and the underlying mechanisms by which DS outperforms other methods remain somewhat unclear to me. I therefore recommend publication after revision.

[1] In Fig.2, the rank of degradation score for all of the 32767 possible combinations of ancestral species are shown. I presume that the score may depend on the initial condition. For example, if a single species community starts with only 10 cells while a four-species community starts with 40 cells (10 from each), naturally the latter would have higher total biomass and thus potentially better degradation performance. This could introduce a bias where species-rich communities appear more effective simply due to larger biomass. Has this effect been controlled for when comparing degradation scores and assigning ranks to all possible species combinations?

[2] What is the composition of the top-ranked (i.e., most optimal) community? What happens if this optimal composition consists of a single species? In such a scenario, the DS method, which emphasizes species diversity, may not be the most effective. Does DS still outperform other methods even when the optimal solution lies in low-diversity or even monoculture communities?

[3] It seems that DS outperforms other methods largely because it supports the persistence of species that are otherwise prone to extinction—by providing them with sufficiently large initial abundances. If this interpretation is correct, I would expect the performance gap between DS and other methods to shrink when the total simulation time is shortened (e.g., by reducing the time step from t=80). Have the authors considered or tested how performance differences vary with the time-scale of selection?

Reviewer #2: The paper presents a model of microbial population dynamics and toxin degradation, with the goal of selecting for communities with the highest degradation. This is an important and interesting topic, the manuscript is well written and the figures are clear. But the proposed “disassembly method” appears very impractical and I have some more specific comments below.

1. My main concern is that the key feature of the disassembly method is “controlling the ecological dynamics […] tightly” (ll. 265). This means that (the most “optimal") communities are continuously restarted with equal abundances of all surviving community members, extinct species are re-introduced in at least some of the new communities (if I understand correctly), and there is some additional random re-shuffling of species (“migration”). Maintenance of diversity in this method is thus “enforced”, and it is not surprising that this allows DS to explore more of the “community” space. But more importantly, the communities found in this way quickly degrade as soon as the tight control is removed, losing the advantage gained by the tight ecological control (ll. 261ff). It is thus very questionable how practically relevant this artificial selection method is, as this would require constant monitoring or frequent resetting of the (unstable) community. The authors also acknowledge the problem of instability in the discussion (ll. 351ff).

This is compounded by other practical concerns: Building the repository of species from the highest scoring communities after each round requires an isolation step. Apart from the practical and labour-intensive (also mentioned in ll. 289ff) problem of how to differentiate and isolate the different species (streak plate assay and overnight cultures?), this step presents additional selection pressures unrelated to the task of toxin degradation. First and foremost, in practice this step would probably always require all relevant species to be able to grow in isolation, if only for them to grow up to the necessary densities after isolating single/a few clonal cells. This would seem to preclude the selection/evolution of any sort of cross-feeding or division of labour strategies, where for example some species predominantly invest into toxin degradation while relying on the community for at least some metabolic processes. The presented model does not explicitly include the isolation step or cross-feeding interactions, which is fine for simplicity reasons. But the isolation step is an integral part of the DS method and its potential to actively select against a tighter integration of the community and its function needs to be discussed in some more detail. In particular, because the propagule and migrant pool selection protocols do not require an isolation step and thus do not run into the same potential problem.

2. From a more theoretical point of view I have the feeling that as soon as you start isolating species/strains, there are a myriad ways of recombining them for the next round. The proposed DS method is one of those methods that seems to perform better (in some sense) than PS and MS propagation, but why look at this specific method with these specific parameters in the first place? Is there a rational behind this, is it the simplest starting point? More concretely: The authors state that “in the disassembly method, more and better communities can be found by randomly adding and removing species” (ll. 207ff). Maybe I missed this, but is the case of NOT adding and removing species explored (DS without migration)? In other words, how much do the results change if the step of random loss or gain of a species in a subset of communities (ll. 129-130) does not happen? Or if the number of randomly chosen communities (here 5) varies (briefly touched upon in ll. 304, but not discussed?)? I guess this corresponds to changing the speed of exploration of the “community space” in some sense? This would be interesting, especially since the PS and MS methods (no emigration or immigration) are compared to corresponding variants with migration (PIS and MIS, ll. 205ff). In general I think the section exploring different values for some of the key parameters (ll. 287ff) would benefit from some more detailed explanations or discussions. For example, it is just mentioned that “the number of communities receiving an invading species is negatively correlated with degradation”. This sounds curious, but what is the reason for or significance of this observation?

3. ll. 82 and ll. 565: Does fixing the initial number of cells per species really correspond to maintaining heritability? While this might ensure persistence of species (which is part of heritability, I guess?), it also resets the final community state of the previous round (i.e. destroying a heritable(?) “trait” of the community). More generally, what does heritability actually mean in this context? Maintaining the presence/absence of species? I think the authors briefly touch upon this in ll. 347, mentioning that they are “sidestepping the issue of heritability”. Here I was also wondering if the suggestion to “achieve individuality first, and selection [for degradation] second” is somewhat arguing against the DS method and more in favor of e.g. the PS method? Since the PS method lets the ecological dynamics stabilize and then selection for degradation could act on the resulting (stable but still “mutating”) community?

4. I was somewhat confused by the paragraph ll. 362 in the discussion. If one removes the isolation and disassembly step, what sort of selection method am I left with? Some form of PS or MS, depending on how I put together the new round of selection?

5. IBM vs. ODE model: If both models show similar results, why present the IBM model and its results? An ODE model seems more straightforward and potentially easier to reproduce? Can the authors give a rational for why to present the IBM?

**Have the authors made all data and (if applicable) computational code underlying the findings in their manuscript fully available?**

Reviewer #1: Yes

Reviewer #2: Yes

PLOS authors have the option to publish the peer review history of their article (what does this mean?). If published, this will include your full peer review and any attached files.

Reviewer #1: No

Reviewer #2: No

**Figure resubmission:**
---

## [Decision Letter · Decision Letter 1]

9 Dec 2025

PCOMPBIOL-D-25-01028R1

Novel artificial selection method improves function of simulated microbial communities

PLOS Computational Biology

Dear Dr. Mitri,

Thank you for submitting your manuscript to PLOS Computational Biology. After careful consideration, we feel that it has merit but does not fully meet PLOS Computational Biology's publication criteria as it currently stands. Therefore, we invite you to submit a revised version of the manuscript that addresses the points raised during the review process.

We look forward to receiving your revised manuscript.

Kind regards,

Chaitanya S Gokhale, PhD

Academic Editor

PLOS Computational Biology

Tobias Bollenbach

Section Editor

PLOS Computational Biology

**Additional Editor Comments:**

Der Authors,

As you can see fro mthe comments, you have resolved the queries of the reviewers very much in detail.

As far as the comments of Reviewer 2 go, I tend to agree. Especially with the introduction of the ODE model i thought it would take more of a a center stage.

The mean field model would not be precise enough to capture all the details of the disassembly method but has the potential to approximate it in specific limits.

That I firmly believe has the benefit of providing understanding over quantitative predictability again not specifying the exactness of the various processes that the Reviewer 2 mentions.

An idea that resonates in the following work - (https://journals.asm.org/doi/10.1128/msystems.00929-22)

Given that the paper is in Computational Biology and the focus on the specific details of the chosen method, I would recommend to highlight that as the reason to choose the IBM over the ODE system.

One could then refer to the mean field as a limiting case.

i think this rephrasing etc would not need to be too extensive.

In general, otherwise I believe that this is an exciting contribution to the journal and am very much looking forward to seeing it out in print!

Thank you

**Journal Requirements:**

1) We note that your Figures files are duplicated on your submission. Please remove any unnecessary files from your revision, and make sure that only those relevant to the current version of the manuscript are included.

**Reviewers' comments:**

Reviewer's Responses to Questions

Reviewer #1: The authors have addressed all of my comments satisfactorily, and I now recommend this manuscript for publication.

Reviewer #2: I thank the authors for their detailed responses and am going for "accept" at this point.

I still do not buy into "favoring" the IBM model over the ODE model, at least not based on the arguments the authors have put forward in this specific case. And now there seems to be a little contradiction between "both models showed very similar results" (ll. 72) and the newly added "The ODE model therefore only recapitulates a subset of the results" (ll. 419).

Generally I think the more concise, tractable and reproducible definition of a model in terms of a system of differential equations far outweighs any, often just perceived, drawbacks. The system could probably be written down in just a few lines (even considering the complication of stochastic mutations, discrete transfers/reassemblies and the resulting hybrid dynamical system) and computationally it is no (big) problem to solve ODE systems with hundreds or even thousands of state variable/equations. And arguably this shouldn't ever really be necessary for these kinds of problems anyway (only a (potentially large, but still relatively smallish) subset of all possible species/mutants will ever have non-zero densities over a given timespan; one can reasonably discretize the "mutant space" to reduce the number of all possible mutants; one can "cheat" a bit by removing mutants if densities fall below a certain extinction threshold, i.e. resizing the system dynamically; etc.).

The IBM model on the other hand is effectively "defined" by, as far as I can see, thousands of lines of code. Realistically this makes it impossible, or at least very impractical, to trace what really is going on, even with the (nice!) inline comments and ten page documentation. I'd also be surprised if this is computationally more efficient than numerically solving a (very) large ODE system.

Again, notwithstanding my above "more a comment than a question" kind of comment, I am ok with recommending "accept".

**Have the authors made all data and (if applicable) computational code underlying the findings in their manuscript fully available?**

Reviewer #1: Yes

Reviewer #2: Yes

PLOS authors have the option to publish the peer review history of their article (what does this mean?). If published, this will include your full peer review and any attached files.

Reviewer #1: No

Reviewer #2: No

**Figure resubmission:**
---

## [Editor Report · Decision Letter 2]

22 Dec 2025

Dear Dr. Mitri,

We are pleased to inform you that your manuscript 'Novel artificial selection method improves function of simulated microbial communities' has been provisionally accepted for publication in PLOS Computational Biology.

Best regards,

Chaitanya S Gokhale, PhD

Academic Editor

PLOS Computational Biology

Tobias Bollenbach

Section Editor

PLOS Computational Biology

Dear Sara,

Thanks a lot for looking through the comments and implementing some edits to addresses the remnant queries.

This is a super paper.

Apart from some typesetting issue in referencing the Alg., which can be fixed in the proofs, I am very happy to see this paper through.

Best,

Chaitanya

---

## [Editor Report · Acceptance letter]

PCOMPBIOL-D-25-01028R2

Novel artificial selection method improves function of simulated microbial communities

Dear Dr Mitri,

I am pleased to inform you that your manuscript has been formally accepted for publication in PLOS Computational Biology. Your manuscript is now with our production department and you will be notified of the publication date in due course.

With kind regards,

Judit Kozma
